# Striato-pallidal oscillatory connectivity correlates with symptom severity in dystonia patients

Roxanne Lofredi [1,2], Lucia K. Feldmann [1], Patricia Krause[1], Ute Scheller [1,3], Wolf-Julian Neumann [1,4], Joachim K. Krauss [5], Assel Saryyeva[5], Gerd-Helge Schneider[6], Katharina Faust[6], Tilmann Sander [7] & Andrea A. Kühn [1,4,8,9,10] ✉

Dystonia is a hyperkinetic movement disorder that has been associated with an imbalance towards the direct pathway between striatum and internal pallidum, but the neuronal underpinnings of this abnormal basal ganglia pathway activity remain unknown. Here, we report invasive recordings from ten dystonia patients via deep brain stimulation electrodes that allow for parallel recordings of several basal ganglia nuclei, namely the striatum, external and internal pallidum, that all displayed activity in the low frequency band (3–12 Hz). In addition to a correlation with low-frequency activity in the internal pallidum (R = 0.88, P = 0.001), we demonstrate that dystonic symptoms correlate specifically with low-frequency coupling between striatum and internal pallidum (R = 0.75, P = 0.009). This points towards a pathophysiological role of the direct striato-pallidal pathway in dystonia that is conveyed via coupling in the enhanced low-frequency band. Our study provides a mechanistic insight into the pathophysiology of dystonia by revealing a link between symptom severity and frequency-specific coupling of distinct basal ganglia pathways.

Isolated dystonia is a hyperkinetic movement disorder that is characterized by involuntary muscle contraction causing twisting movements and abnormal postures[1]. Although the underlying disease mechanisms are not conclusively understood, dystonia is considered a network disorder linked to abnormal neuronal inhibition and plasticity resulting in pathological signaling of the circuit between sensorimotor cortices, cerebellum and the basal ganglia (BG)[2,3]. Within the basal ganglia, PET studies demonstrated a specific increase of striatal D1- and a decrease of D2-receptors that was attributed to a potential hyperactivity of the direct pathway between the striatum (STR) and the

globus pallidus internus (GPi)[4,5]. Functional correlates of neuronal activity in dystonia patients providing further evidence for the hypothesis of a hyperactive direct pathway between striatum and internal pallidum are currently missing. The interconnections between basal ganglia nuclei are difficult to assess because recordings of neuronal activity are mainly restricted to single nuclei targeted by therapeutic deep brain stimulation (DBS) in patients with movement disorders. These DBS-target nuclei, such as the GPi, subthalamic nucleus (STN) or ventral intermediate thalamus (VIM), display specific patterns of oscillatory neuronal activity that have been associated with

[1]Department of Neurology, Charité-Universitätsmedizin Berlin, Berlin, Germany. [2]Berlin Institute of Health (BIH), Berlin, Germany. [3]Department of Neurology, Universität Göttingen, Göttingen, Germany. [4]Bernstein Center for Computational Neuroscience, Humboldt-Universität zu Berlin, Berlin, Germany. [5]Department of Neurosurgery, Medizinische Hochschule Hannover, Hannover, Germany. [6]Department of Neurosurgery, Charité-Universitätsmedizin Berlin, Berlin, Germany. [7]Physikalisch Technische Bundesanstalt, Abbestraße 2, Berlin, Germany. [8]NeuroCure, Exzellenzcluster, Charité-Universitätsmedizin Berlin, Berlin, Germany. [9]DZNE, German Center for Neurodegenerative Diseases, Berlin, Germany. [10]Berlin School of Mind and Brain, Humboldt-Universität zu Berlin, Berlin, Germany. ✉e-mail: Andrea.kuehn@charite.de

hyper- or hypokinetic motor states. While hypokinetic symptoms as seen in Parkinson's disease (PD) or as side-effect of pallidal DBS in dystonia have been related to an increase of beta oscillations (13–35 Hz)[6,7], hyperkinetic symptoms of dystonia or Tourette's syndrome have been associated with exaggerated low-frequency oscillations (3–12 Hz)[8–10]. It has been speculated that these oscillatory patterns can be attributed to opponent pathway interactions of the anti-kinetic indirect pathway for beta and the pro-kinetic direct pathway for low-frequency activity. Particularly for the latter, evidence from multisite recordings in striato-pallidal circuits is lacking. If low-frequency activity would reflect direct striato-pallidal pathway activity, one would hypothesize that firing rates in the upstream striatum should correspond to the increased low-frequency oscillatory activity in the downstream GPi, as oscillations are thought to mainly reflect the input to the recorded structure[11]. Only two studies have investigated firing rates from the striatum in dystonia patients, recorded while the microelectrode trajectory was passed to the GPi as final DBS-target[12,13]. One of these studies indeed reported an increase of neuronal firing rates in the striatum up to 9 Hz, which would match the exaggerated low-frequency activity in downstream GPi of dystonia patients as found in other publications. However, the lack of parallel recordings from both areas of the direct striato-pallidal pathway have so far obviated a circuit description of this activity between striatum as the input and GPi as the output hubs of the basal ganglia. Characterizing circuit activity within the direct pathway would strengthen the hypothesis of the pathophysiological role that basal ganglia pathway balance plays in the etiology of dystonia. Here, we make use of specific DBS-electrode types in which contacts span over -15 mm to perform these unique, parallel multi-region recordings of neural population activity from the striatum, as well as the GPi and Globus pallidus externus (GPe) in ten dystonia patients. We localize all electrode contact pairs in standardized space to assign them to the respective structures and validate their connectivity profile within the motor circuit using normative connectomes. We characterize spectral features and connectivity profiles for all structures in the canonical low-frequency (3–12 Hz) and beta band (13–35 Hz) and investigate their specific link to dystonic symptom severity for direct (STR-GPi) and indirect (STR-GPe) BG pathway. We are thereby able to capture neural circuit dynamics of direct and indirect basal ganglia pathways and identify their pathophysiological relevance in dystonia patients.

## Results

### Localization of basal ganglia recording sites
Electrode reconstructions and MRI network analysis (Fig. 1) confirmed that recording sites resided in sensorimotor circuits. Striatal recording sites were localized in the posterior putamen, i.e., the sensorimotor domain of the striatum[14,15] (mean MNI coordinates across hemispheres: x = ±21.8 ± 1.5, y = −1.1 ± 1.5, z = 6.4 ± 1.4). In the GPi and GPe, recording sites were distributed across the posterior section of the pallidum (GPi: x = 20.9 ± 1.2, y = −7.5 ± 1.4, z = −4.4 ± 1.5; GPe: x = ±21.5 ± 1.2; y = −3.9 ± 1.5; z = 1.3 ± 1.5). The averaged functional connectivity profile of recording sites in the striatum revealed positive coupling with primary motor cortex, premotor areas and cerebellum, confirming a sensorimotor location of the DBS-recording sites within the posterior putamen.

### Spectral pattern of striatal and pallidal oscillatory activity
We found a similar pattern of neuronal activity in the three basal ganglia nuclei in all patients. Distinct spectral peaks in the low-frequency range were present in all striatal, all GPi and 17 out of 19 GPe recordings with mean peak frequencies of about 8.5 Hz across structures and patients (ranging from 3 to 12 Hz). Figure 2 shows the mean peak frequency for striatum: 8.5 ± 2.5 Hz; GPi: 8.4 ± 2.9 Hz and GPe: 8.7 ± 2.1 Hz across the 19 hemispheres. Across nuclei, spectral peaks in the beta frequency range were less frequent and detected in 13 out of

19 hemispheres with peak frequencies ranging from 13 to 35 Hz (STR: 23.5 ± 6.8 Hz; GPi: 21.7 ± 6.6 Hz; GPe: 23.4 ± 6.9. Hz). There was no significant difference in peak frequency across structures, neither for the low-frequency nor the beta band (P > 0.05). Averaged spectral power of the low-frequency and beta band was similar across all structures (low-frequency band: STR = 5.8 ± 1.7 a.u., GPi = 5.5 ± 1.1 a.u., GPe = 5.6 ± 1.6 a.u.; beta-band: STR = 0.6 ± 0.5 a.u., GPi = 0.4 ± 0.2 a.u., GPe = 0.5 ± 0.4 a.u.). There was no significant difference in spectral power or peak frequency in the LF-band, when comparing cervical (6 patients, 11 hemispheres) with segmental or generalized dystonia (4 patients, 8 hemispheres), see Supplementary Table 2.

### Functional connectivity across basal ganglia structures
Imaginary part of coherence revealed significant coupling over the averaged low-frequency band across all structures when compared to shuffled data (see Fig. 3; STR-GPi: iCOH = 0.07 ± 0.02, shuffled = 0.02 ± 0.006, P = 0.002; STR-GPe: iCOH = 0.066 ± 0.027, shuffled = 0.03 ± 0.0096, P = 0.003; GPi-GPe: iCOH = 0.067 ± 0.034, shuffled = 0.02 ± 0.009, P = 0.003), which showed no significant difference between cervical (6 patients) and segmental or generalized dystonia (4 patients), see Supplementary Table 2. These findings could be reproduced by using the weighted phase-lag-index (wPLI) that showed significantly higher coupling in the low-frequency band when compared to shuffled data (see Supplementary Fig. 3; STR-GPi: wPLI = 0.14 ± 0.05, shuffled = −0.001 ± 0.01, P = 0.002; STR-GPe: wPLI = 0.16 ± 0.13, shuffled = 0.0049 ± 0.01, P = 0.004; GPi-GPe: wPLI = 0.14 ± 0.1, shuffled=0.00001 ± 0.01, P = 0.002). Granger causality without time-reversed correction peaked at 5 Hz both with the striatum and the GPi as source. There was no significant difference between time-reversed granger causality from either source compared to the other (striatum to GPi = 0.0041 ± 0.04; GPi to striatum = 0.0014 ± 0.01, P > 0.05), suggesting net bidirectional information transfer when averaged over the low-frequency band. When tested for each frequency bin within the predefined low-frequency range separately, striatum drove the largest parts of the low-frequency band in the GPi in 12 of 19 hemispheres.

### Correlation of low frequency activity and symptom severity.
As shown in Fig. 3, we observed a significant positive correlation between dystonic symptom severity and low-frequency power in the GPi as the main output nucleus of the basal ganglia (R = 0.89, P = 0.0014), but not the striatum as their main input nucleus nor the GPe. In addition to low-frequency power in the GPi, the coupling strength as measured by imaginary part of coherence in the low-frequency band between striatum and GPi – which presumably reflects direct pathway activity - correlated significantly with symptom severity (R = 0.74, P = 0.009), see Fig. 3F. In contrast, low-frequency coupling strength between striatum and GPe – as possible correlate of indirect pathway activity - did not (R = −0.002, P = 0.13, Fig. 3F), nor did coupling strength between GPi and GPe (R = 0.6, P = 0.05). Averaged power or coupling in the beta band did not correlate with symptom severity.

## Discussion
In this study, we report unique, parallel recordings of oscillatory activity in the striatum, GPi and GPe of dystonia patients. In all recorded basal ganglia nuclei, low-frequency oscillations were detected, but only in the GPi their power was significantly linked to symptom severity. In addition, we provide neurophysiological evidence for an association between dystonia pathophysiology and striato-pallidal pathway activity by revealing that the coupling strength in the low-frequency band, specifically between striatum and GPi, correlates with symptom severity.

These results integrate into a large body of evidence suggesting that altered patterns of synchronization in neuronal assemblies as recorded by oscillatory activity may underlie pro- and anti-kinetic

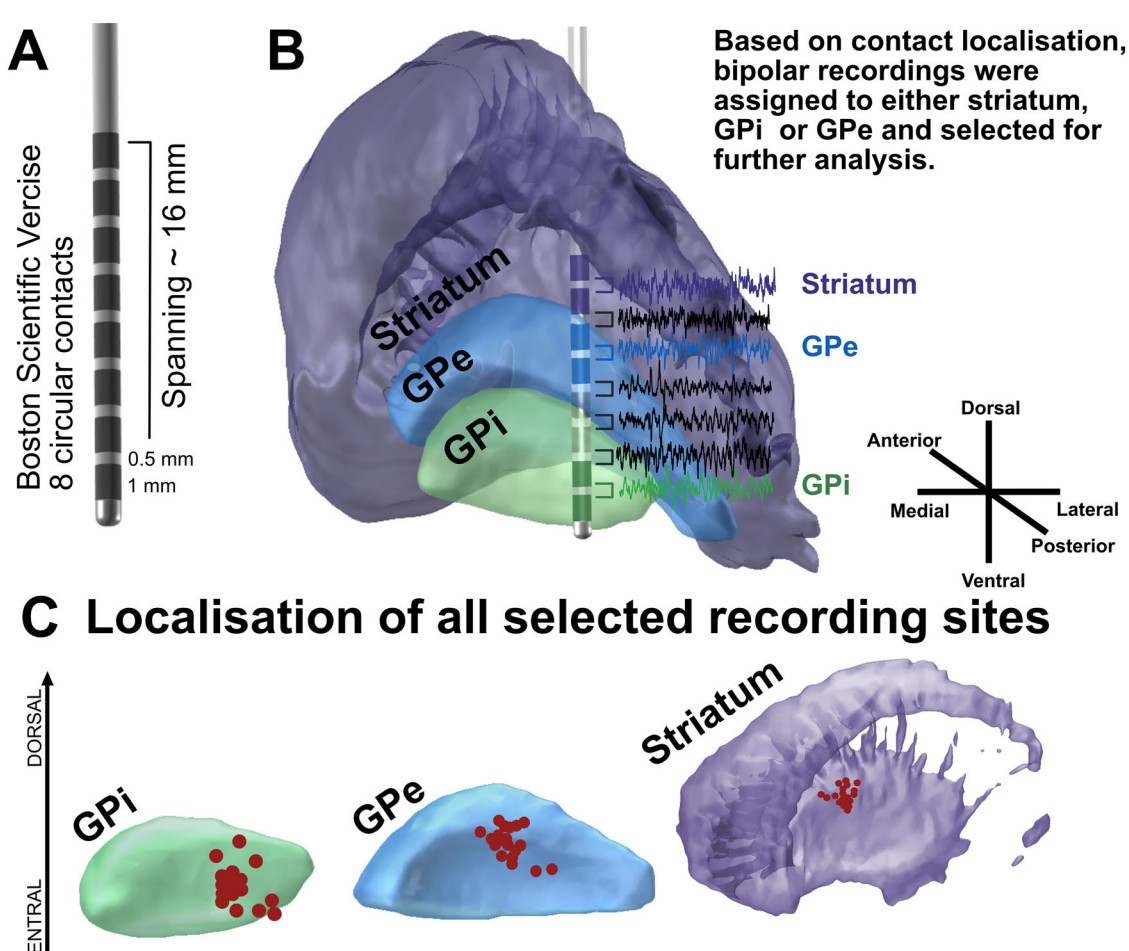

**Fig. 1 | Methodological set-up and recording sites across basal ganglia nuclei of all patients. A** In this study, the DBS-electrode Boston Scientific Vercise was implanted in all patients. Note that this DBS-electrode model is composed of eight circular contacts that span over 15.5 mm. **B** Shown is an exemplary recording in the post-operative period. In all patients, we performed bipolar recordings from adjacent contacts resulting in 7 parallel recordings per hemisphere. After DBS-electrode localization, recordings were assigned to either striatum, GPi or GPe if two recording contacts were localized in the respective structure. In the shown example, this would fit to the recording between contact 1 and 2, which are both localized in the GPi (green); contact 5 and 6, both localized within the GPe (blue) and contact 7 and 8, both localized within the striatum (purple). Thus, these recordings were retained for further analysis. In contrast, all other parallel recordings (namely bipolar recordings from contact 2 and 3, contact 3 and 4 as well as contact 6 and 7 as colored in black) are removed from further analysis steps. Thereby, only three parallel recordings per hemisphere (striatum, GPe and GPi) are kept from the original seven. **C** Shown are the exact localization within striatum (purple), GPe (blue) and GPi (green) of these retained recording sites from all hemispheres across all patients (red dots). For visualization purposes all recording sites have been flipped to the respective nuclei of the right hemisphere.

signaling within basal ganglia pathways[16,17]. Locally, exaggerated synchronization in beta oscillations has been linked to the occurrence of hypokinetic symptoms[6,7,18], while low-frequency and gamma oscillations have been associated with voluntary and involuntary movements across movement disorders[8,10,19–23] and thus been qualified as pro-kinetic. Due to technical constraints, previous studies were restricted to recording sites within single nuclei of the basal ganglia, mostly DBS-target structures such as the GPi or STN. Importantly, our study connects these local observations to coupling patterns across the basal ganglia, as parallel recordings of several basal ganglia nuclei, most importantly including the striatum, have been lacking in humans.

Here, we show that low-frequency oscillations, as a neural population read-out of pro-kinetic signaling, can be detected in the striatum, the input structure of the basal ganglia loop. These oscillations may arise from pathological circuit interactions of cortico-striatal and basal ganglia pathways in dystonia, which can rhythmically modulate excitability and firing probability of individual neurons within the striatum as previously observed[13]. Dystonia is increasingly understood as a circuit disorder and many brain areas beyond the basal ganglia

have been implicated in its' pathophysiology, such as (pre-)motor and sensory cortices[24] as well as the cerebellum[25,26]. Thus, it is unclear, whether the observed abnormal neuronal pattern are spreading from these brain areas via the extensive connectivity of striatum or are of local origin within the basal ganglia. Here, we report oscillatory activity patterns that are thought to reflect afferent synaptic voltage fluctuations[11]. Thus, if pathological low-frequency signaling would predominantly arise upstream to the basal ganglia, oscillatory power in the striatum, as input nucleus of basal ganglia, should already show a link to dystonic symptom severity. The observation that the link between low-frequency power and dystonic symptoms is lacking in the striatum and specifically found in the GPi and for connectivity between direct striatopallidal pathway but not indirect striatum-GPe or GPe-GPi pathway, may thus be interpreted as hint toward a basal ganglia source of this pathological phenomenon. Still, follow-up studies with parallel electrocorticography (ECoG) or whole-head M/EEG recordings and basal ganglia activity could reveal additional mechanisms on the interplay between pathophysiological basal ganglia activity and other hubs of the sensorimotor circuit.

## A Power spectra aligned on peak frequency

## B Peak frequencies across patients

**Fig. 2 | Spectral patterns are similar across basal ganglia nuclei in dystonia patients. A** When averaged across hemispheres of all subjects, similar spectral patterns are observed with spectral peaks in the low-frequency (upper row) and beta band (lower row) in the striatum (STR, left column), GPi (mid column) and GPe (right column). Averaged power spectra across hemispheres and patients are colored (STR=purple, GPi=green, GPe=blue), while individual power spectra of each hemisphere are plotted in grey. For visualization purposes, power spectra are flattened and aligned to the individual peak frequency. No significant difference in power spectral density at peak frequency is seen across basal ganglia nuclei. **B** Similarly, peak frequencies in the respective frequency bands (low frequency and beta band) in the 19 hemispheres per structure did not significantly differ across basal ganglia nuclei and were distributed across the low frequency (upper row) and beta band (lower row) as validated by permutation tests and FDR-corrected for multiple comparisons. n.s. not significant. In box plots, the whiskers indicate minimal and maximal values per hemisphere, the central marks indicate the median and edges the 25th and 75th percentiles of the distribution. A.u. arbitrary units.

Importantly, the correlation with dystonic symptom severity observed with pallidal low-frequency power could be extended to striato-pallidal coherence. As coherence is thought to underlie effective and selective neuronal communication across brain areas[24], this finding is indicative of pathological alterations in the coupling of basal ganglia input and output hubs. In this framework, low-frequency activity may be transformed or amplified in striato-pallidal synapses to reach an excessive pathological synchrony that impairs motor circuit activity and drives involuntary muscle contractions in dystonia. Notably, this correlation was specifically strong in the connection between striatum and GPi that form the direct pathway and absent for connectivity between striatum and GPe that form the indirect pathway. Consequently, we may speculate that exaggerated striato-pallidal coherence may relate to excessive impact of D1 expressing direct pathway medium spiny neurons on the GPi, that were previously reported to be pathologically upregulated in dystonia[4,5]. In the future, cell-type specific recordings in human patients or animal models of dystonia may shed further light on this speculation.

Either way, low-frequency oscillations seem not confined to the direct pathway, as they are also present in nuclei of the indirect pathway such as the GPe – albeit less prominent[27]. Previous studies in dystonia patients have also reported low-frequency synchronization in the STN, another strategic nucleus of the indirect pathway[25,26].

However, a correlation between low-frequency oscillations in nuclei of the indirect pathway and dystonic symptom severity has not been reported yet and did not reach significance in the present study. In the same vein, the suppression of LF activity with DBS, which has been described for pallidal DBS[28], seems not to be the primary mechanism of action of DBS that targets nuclei of the indirect pathway: A recent study investigating the effects of subthalamic DBS, which is also an effective target nuclei for the treatment of dystonia, reported a suppression of beta instead of LF activity[29]. Thus, low-frequency oscillations may spread across bidirectional connections to the indirect pathway, while the critical vulnerability for symptom generation resides in striato-pallidal interactions. Notably, our results have been obtained in the acute perioperative period, where surgical intervention and patient state may have affected neural recordings, e.g., through the so-called microlesion effect. With the rise of new sensing enabled devices, it will be important to extend these findings to dynamic network changes with chronic DBS, alongside the symptom alleviation which can take up to months after implantation.

We conclude that low-frequency oscillations are a ubiquitous signature of basal ganglia activity in dystonia patients, but the association with impaired motor signaling is specific for the functional coupling between the striatum and GPi, possibly reflecting direct pathway activity. These findings provide a functional correlate of neuronal activity in dystonia patients that strengthen the hypothesis of an imbalanced striato-pallidal pathway activity as crucial hallmark in the pathophysiology of dystonia.

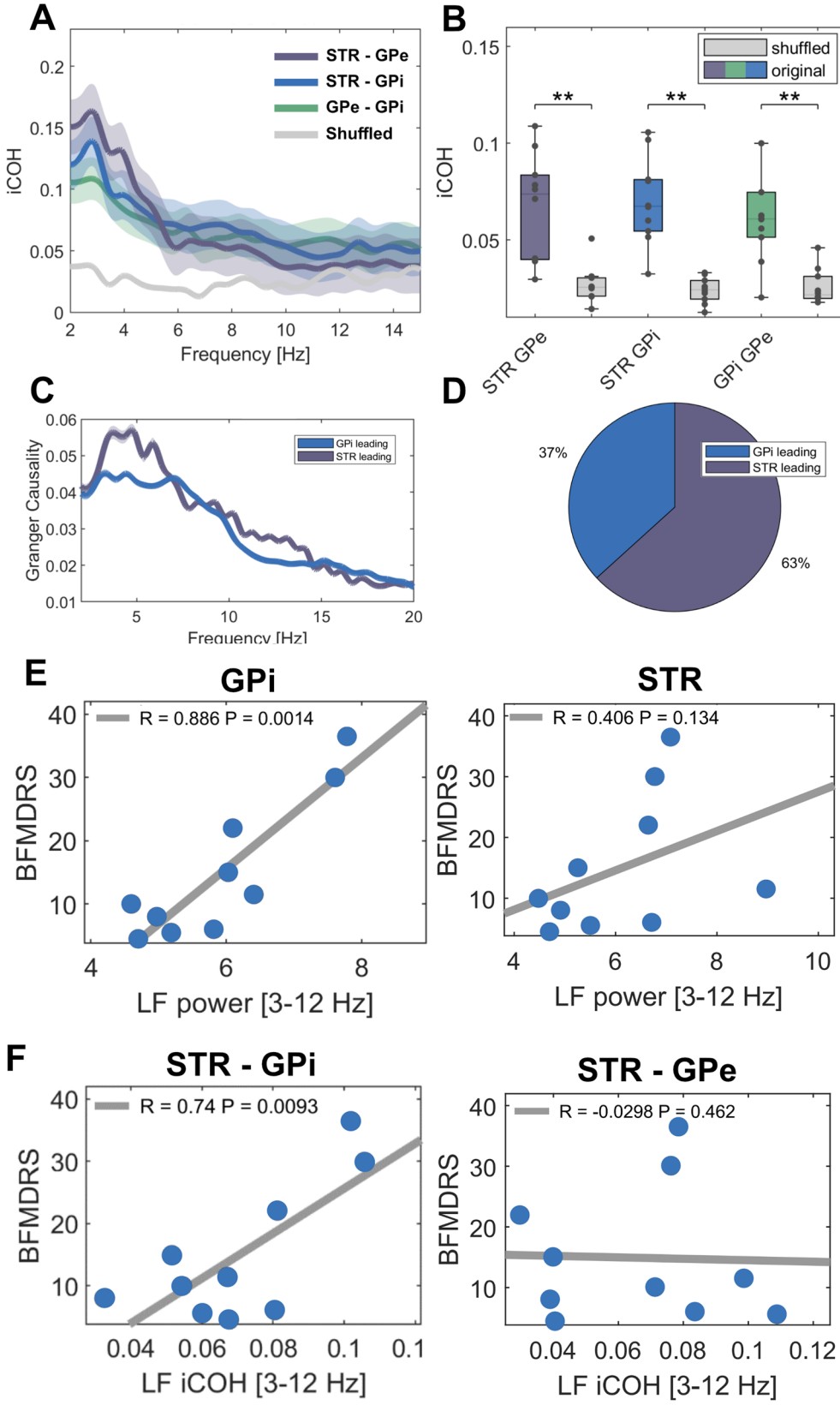

These findings from the acute, post-operative period may inspire the development of neurotechnological devices that allow for chronic recordings of several basal ganglia nuclei in order to develop a circuit-based approach to neuromodulation therapies and incorporate pathophysiological network signatures as feedback signals for future and demand-adapted treatment strategies.

## Methods

### Patients, surgery and DBS-electrode localization

We included 10 patients undergoing pallidal DBS for cervical, segmental or generalized dystonia (5 male, 5 female, aged 51,2 ± 11,4 years) at Charité-Universitätsmedizin Berlin ($n = 7$) or Medizinische Hochschule Hannover ($n = 3$). In all patients, dystonic symptom severity was

**Fig. 3 | Connectivity patterns across basal ganglia structures and their correlation to symptom severity. A** Shown are averaged coherence spectra (iCOH= imaginary part of coherence) between striatum and GPe (purple), striatum and GPe (blue), GPe and GPi (green) and averaged shuffled coherence (grey) across hemispheres. Shaded areas indicate standard deviation of the mean. **B** All recorded basal ganglia structures are functionally coupled in the low-frequency band as shown by significantly higher coherence values (colored box plots) when compared to shuffled data (grey box plots); STR-GPi: $P = 0.002$; STR-GPe: $P = 0.003$; GPi-GPe: $P = 0.003$ as validated by permutation tests and FDR-corrected for multiple comparisons. Black dots indicate mean value of coherence per hemisphere ($n = 19$ per basal ganglia structure). In box plots, the whiskers indicate minimal and maximal values per hemisphere, the central marks indicate the median and edges the 25th and 75th percentiles of the distribution. **C** Shown are averaged spectra of Granger causality with GPi (blue) or striatum (purple) as source. Shaded areas indicated standard deviation of the mean. **D** When each frequency bin is considered separately, the majority of the low-frequency band is led by the striatum in 63% and by the GPi in 37% of cases. **E** Dystonic symptom severity correlates with averaged low-frequency power in the GPi, but not the striatum. **F** Moreover, the significant coupling strength of low-frequency activity between the striatum and the GPi, but not between striatum and GPe hints towards an involvement of the direct but not the indirect pathway. **E**, **F** show Pearson's correlation coefficients. ** $p < 0.01$.

assessed using the Burke–Fahn–Marsden Dystonia Rating Scale (BFMDRS). For clinical details, see Table 1. All patients gave written informed consent, approved by the local ethics committee of Charité – Universitätsmedizin Berlin and Medizinische Hochschule Hannover in accordance with the standards set by the Declaration of Helsinki. The DBS-electrode model used was Vercise™ DBS, DB-2201 (Boston Scientific), with 8 circular contacts spanning over 15.5 mm, see Fig. 1A. DBS-electrodes were localized by fusing pre- and postoperative imaging and transferred to Montreal Neurological Imaging (MNI)-space with the default pipeline of the Lead-DBS toolbox[30] that has been validated as robust against normalization to standardized spaces[31] and interrater variance[32], see Fig. 1B. DBS-electrodes showed at least two contacts within the striatum, two contacts within the GPe and two contacts within the GPi (see Supplementary Fig. 1 and Supplementary Table 1), according to segmentation of basal ganglia nuclei in the DISTAL atlas[33], an exemplary case is shown in Fig. 1B. Of the 19 patients with dystonia that underwent DBS surgery in our centers, 10 patients fulfilled these pre-established inclusion criteria and were thus considered for further analysis of the neurophysiological data. In patient #7, no contact of the left DBS-electrode was localized in the striatum and the left hemisphere was thus excluded, resulting in 19 hemispheres included in the present study. In one case (patient # 6, right hemisphere), one DBS-contact of the GPi-recording was placed slightly ventral to the GPi, in adjacent white matter. Primary DBS target was the posteroventral lateral, motor portion of the GPi. Further details on surgical planning as well as electrode localization in individual MRI-sequences can be found in the supplementary material, Supplementary Table 3 and Supplementary Fig. 4. The trajectory of DBS-electrode insertion led to similar, yet slightly more anterior recording sites in DBS-electrode contacts localized in the posteroventral striatum. To assure their localization within the motor circuit, we conducted a connectomic analysis of the patient individual recording sites within the respective basal ganglia nucleus, shown in Supplementary Fig. 3. Whole-brain connectivity in an openly available group connectome was estimated by using a 2 mm seed roughly reflective of the neural field recorded from DBS contacts (for more details regarding this approach see Supplementary Materials). This analysis provides additional evidence for the recording sites being part of the sensorimotor circuit within the basal ganglia but does not enable the investigation of symptom-specific changes in neuroimaging-based connectivity.

## Recordings

Three minutes of rest activity was recorded bipolarly from adjacent contacts in the 1–7 days following surgery while DBS-leads were still externalized. The implanted DBS-electrode (Boston Scientific Vercise™ DBS) is composed of eight circular contacts spanning over 15.5 mm, thereby allowing parallel bipolar recordings of three basal ganglia nuclei (namely the striatum, GPe and GPi). However, due to the electrode design and pulse generator type no directional steering nor chronic neuronal recordings are available. Signals were amplified (50.000x) and recorded at a sampling frequency of 1 kHz either using a D360 amplifier (Digitimer, Hertfordshire, UK) and a 1401 A-D converter (CED, Cambridge, UK) onto a computer using Spike2 software ($n = 8$)

or through the integrated EEG-system of a 125 channel MEG system (YOKOGAWA ET 160), $n = 2$.

## Signal processing

Artifact-free recording segments of $210 \pm 105$ s were analyzed from the contact pairs assigned to the region of interest (i.e., striatum, GPi, GPe) following DBS-electrode localization. Signal analysis was performed using custom MATLAB code, based on SPM12[34] and FieldTrip[35]. Continuous recordings were down sampled (200 Hz), filtered (high pass = 1 Hz, low pass = 98 Hz and notch filter = 48–52 Hz) and transferred to the frequency domain using Morlet wavelets. For power analyses, spectra were normalized to the standard deviation, which corrects for differences in electrode impedance, and averaged over the following canonical frequency bands: low-frequency (3–12 Hz) and beta band (13–35 Hz) as both have been shown to correlate with hyper- or hypokinetic symptoms in dystonia[7,8]. Functional connectivity between basal ganglia nuclei was assessed by the imaginary part of coherence, which provides a frequency domain measure of linear phase and amplitude relationships between signals, while removing instantaneous interactions[36]. Coherence measures the correlation between two signals at a given frequency, a complex quantity, which can be broken down into real and imaginary parts. The latter part reflects connectivity without zero phase lag, indicative of true communication of distinct oscillators, without contamination from volume conduction. For our analysis, we thus relied on this more conservative and robust connectivity measure. Given that both complex and absolute measures are commonly called "coherence", we follow the previously suggested terminology and refer to "imaginary part of coherence" in the present text. In addition, we calculated the weighted phase-lag-index as supplementary measure for neuronal coupling, to make sure that the reported results did not rely on false positive coherency[37]. To investigate the directionality of neuronal population activity, we used a non-parametric variant of Granger causality and subtracted the time-reversed data as this suppresses the influence of data asymmetries unrelated to time-lagged interactions[38]. To further describe directionality of the low-frequency band in the absence of net group effects, we assessed the number of frequency bins within the low-frequency band in which the striatum exhibits greater drive than the pallidum, and subsequently designated the hemisphere as possessing overall striatal driving if this condition was met.

## Statistics

Nonparametric Monte Carlo permutation tests were used for across subject comparison. For correlative analyses, Pearson's correlation was calculated after testing for normality using the Kolmogorov-Smirnov-Lilliefors-Test implementation in MATLAB. All results are indicated as mean ± standard deviation and reported significant at an α level of .05. Multiple comparisons were controlled by using detection of the false discovery rate (FDR) in independent or cluster correction in dependent comparisons. For correlative analyses, neurophysiological estimates were averaged across hemispheres to determine a single value per subject as the clinical score (BFMDRS) allows only for a global estimate of symptom severity. No statistical method was used to

**Table 1 | Clinical details**

| No | Sex | Diagnosis | Medication | BFMDRS |
|----|-----|-----------|------------|--------|
| 1 | F | Segmental Dystonia | Zonisamid 50 mg | 15 |
| 2 | M | Cervical Dystonia | none | 8 |
| 3 | F | Segmental Dystonia | Trimipramin 65 mg | 10 |
| 4 | M | Cervical Dystonia | none | 5.5 |
| 5 | M | Generalized Dystonia | none | 11.5 |
| 6 | F | Cervical Dystonia | none | 4.5 |
| 7 | F | Generalized Dystonia | none | 36.5 |
| 8 | M | Generalized Dystonia | Tizanidin 3 mg<br>Mirtazapin 30 mg<br>Valoxan 50 mg<br>Rivotril 0.25 mg<br>Thiamazol 5 mg | 22 |
| 9 | F | Generalized Dystonia | Mirtazapin 15 mg<br>Pregabalin 200 mg | 30 |
| 10 | M | Cervical Dystonia | None | 6 |

predetermine sample size but sample size was instead determined based on previous studies and expected statistical power[7,9,10]. Due to the small sample size number, no sex or gender analysis was performed, but the sample contained an equal number of male and female participants.

**Reporting summary**

Further information on research design is available in the Nature Portfolio Reporting Summary linked to this article.

## Data availability

Data can be made available conditionally to data sharing agreements in accordance with data privacy statements signed by the patients within the legal framework of the General Data Protection Regulation of the European Union. Requests should be directed to the lead contact, Andrea A. Kühn (andrea.kuehn@charite.de), or the Open Data officer (opendata-neuromodulation@charite.de). Source data of figures are provided with this paper. Source data are provided with this paper.

## Code availability

Custom MATLAB code is available online[39] and in the following GitHub repository: https://github.com/neuromodulation/wjn_toolbox/.

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

## Acknowledgements

We thank all patients for their participation as well as Stefan Haufe and Maria del Carmen Herrojo Ruiz for their methodological support. Dr. Lofredi is participant in the BIH Charité Clinician Scientist Program funded by the Charité – Universitätsmedizin Berlin, and the Berlin Institute of Health at Charité (BIH). The work was supported by Deutsche Forschungsgemeinschaft (Project-ID 424778381 – TRR 295 Retune) and European Union's Horizon 2020 research and innovation program EJP RD COFUND-EJP N° 825575 (EurDyscover).

## Author contributions

R.L. and A.A.K. planned and oversaw all aspects of the study. G.H.S., K.F., J.K.K., and A.S. performed the implantation of DBS-electrodes. R.L., L.K.F., U.S., T.S., and W.J.N. performed the intracranial recordings. R.L. and P.K. performed the clinical assessment of symptom severity. R.L. performed localization of DBS-electrodes and connectivity analyses. R.L. and W.J.N. analyzed the intracranial data. R.L. wrote the manuscript with input and substantial revisions from all authors.

## Funding

## Competing interests

Dr. Lofredi, Dr. Feldmann and Prof. Neumann report personal fees from Medtronic. Prof. Kühn reports personal fees from Medtronic, and Boston Scientific. Dr. Krause reports personal fees from Medtronic, Stadapharm and Abbvie. Prof. Krauss reports personal fees from Medtronic, personal fees from Boston Scientific. Dr. Schneider reports personal fees from Medtronic, personal fees from Boston Scientific, personal fees from Abbott. All personal fees are not linked to the here presented study results. The remaining authors declare no competing interests.
