## [Peer Review File · Nature Communications]

Striato-pallidal oscillatory connectivity correlates with symptom severity in dystonia patientsReviewers' comments:

Reviewer #1 (Remarks to the Author):

Dystonia is a hyperkinetic movement disorder arising from functional and structural abnormalities in the basal-ganglia-cortical loops. Abnormal oscillations have been newly repeated postulated but the etiology of this atypical activity in basal ganglia pathways and its connection to dystonic symptoms are currently uncompletley elucidated. This work made use of invasive recordings using deep brain stimulation electrodes from ten dystonia patients showing low-frequency direct striato-pallidal pathway activity, indicating a possible pathophysiological role for this pathway. Dystonic symptom severity correlates with frequency-specific coupling pathways. This study presents original mechanical insight into the pathophysiology of Dystonia by exposing a link between symptom severity and frequency-sensitive coupling.

1) The figures for electrode localization is of bad quality and make anatomical assumptions about the localisaiton of the active contacts impossible. The visualization of the electrode is improper and only schematic and makes also implications on active contacts / directionality / levels where the recordings were applied impossible. This is generally an important point (s. also P4) as also the generalizability of the anatomical specificity is not clear to me. A exact presentation of the each electrode position (not group data) is needed to check the differential localization and the nature of the effects.

2) The general presentation of the results is counterintuitive. Although I appreciate the need for the presentation of single cases for the visualization – the group results are needed to address the main results of the study. The presentation of the results in the Fig 2 (B, C, D) is not comprehensive. The significant contrast are not highlighted and visually relatable.

3) One of the main concerns is a possible acute nature of the effects. With the insertion of the electrodes a microlesion effect occurs and leads to an acute reorganization of the brain networks. The activity in the internal pallidum and also the synchronization between striatum and internal pallidum could be related to symptom improvement or the microlesional effect. The authors should exclude this mechanistic explanation with data from chronically implanted patients A further analyses that could bring an improved causality proof of the results would be the analysis of responders' vs non-responders (i.e. also from the postoperative acutely implanted patients with the micro-lesional effect)

4) A further major drawback is the regional / anatomical specificity of the effects. The synchronization could be related to the influence of a distant pacemaker (i.e. sensory cortex) that sends then a different input to striatum / GPi/ GPe structures..

5) Further I am relatively concerned about the novelty of the work. A rathe vast literature exist on the role of low-oscillation for the generation of dystonic symptoms and their relation to therapeutic responses to DBS (i.e. see <https://pubmed.ncbi.nlm.nih.gov/34328685/>
<https://pubmed.ncbi.nlm.nih.gov/37268239/>
<https://pubmed.ncbi.nlm.nih.gov/30440096/>

A more in-depth discussion of the novelty and pathophysiological clinical implications is needed.

6) The use / relevance of the connectomic analyses are not clear to me. The work makes assumptions to the interconnectivity to the addressed network nodes but no intra-subject demonstration of the network effects with the applied methods is possible. The use of the “generalistic” vs individual connectome is a major drawback. Furthermore this a group of dystonia patients, where no work was done (to my best knowledge) on the comparability of the rs-fMRI between the group and the interapplicability.

7) The authors should comment on the concatenated inclusion of cervical and generalized dystonia patients and the relevance of the results. Do the authors see any differences among these groups?

8) The use of abbreviations is not conclusive (STR is not introduced)

9) „Imaginary“ part of coherence – is misleading

10) The relevance of the MEG recordings / analyses is not clear to me and should be excluded (or better elaborated)

Reviewer #2 (Remarks to the Author):

This paper looked into the neurophysiological coupling among the striatum, the external and the internal pallidum with unique simultaneously recorded local field potentials. The research provides insights into the functional coupling within direct striato-pallidal pathway in dystonia. The study confirmed previous findings that the low frequency oscillations in GPi are correlated to the dystonic symptom severity. More interestingly, dystonic symptoms correlate specifically with low-frequency coupling between striatum and internal pallidum.

The local field potentials were recorded from adjacent contacts and there may be volume conduction within such local region. The imaginary part of coherence was used to reduce the influence of volume conduction, but it may cause false positive coherency (Ghost interactions in MEG/EEG source space: A note of caution on inter-areal coupling measures.

DOI10.1016/j.neuroimage.2018.02.032). It would be essential to be cautious to such measures and exclude the spurious interaction. It might be useful to use another measure, for instance, weighted phase lag index, to confirm the findings.

Reviewer #3 (Remarks to the Author):

The authors investigated how indirect and direct pathways of the basal ganglia circuits are involved in the pathophysiology of dystonia. This is the first study in humans analyzing parallel measurements from the Striatum, the GPe, and the GPi. The authors examined 20 patients and analyzed 19 hemispheres. With Boston Vercise standard leads, they placed two ring contacts with one insertion to the three targets. They measured signals from the contact pairs in a bipolar

configuration. Functional connectivity with motor cortex areas and the cerebellum confirmed the correct electrode locations.

The working group used an innovative method to map the direct and indirect basal ganglia network activity. The results are consistent with previous literature data.

The results:

They found a 3-12Hz activity and a less frequent 13-35Hz activity in all three targets without a difference in the peak frequency or power across the structures.

They revealed low-frequency, bidirectional information flow between the Striatum and the GPi, in which the Striatum dominated in more cases.

They found a significant positive correlation between the low-frequency power in the GPi, the low-frequency coupling of the Striatum and GPi, and the severity of dystonia.

This explorative study has an appropriate number of subjects. The study planning and the method used are exact. The statistical analysis is well performed. The presentation of the results is logically structured, and the Results paragraph is easy to follow. The whole article is enjoyable to read; the essential results are adequately emphasized in the text.

Comments:

1. In the legend of Fig 3A, please define the light-color intervals.
2. The last two sentences of the Fig 3 legend should relocate to the legend of Fig 3B.

In summary, this is an excellent and innovative work. I recommend publishing the manuscript in the journal.

Reviewer #1:

Dystonia is a hyperkinetic movement disorder arising from functional and structural abnormalities in the basal-ganglia-cortical loops. Abnormal oscillations have been newly repeated postulated but the etiology of this atypical activity in basal ganglia pathways and its connection to dystonic symptoms are currently uncompletely elucidated. This work made use of invasive recordings using deep brain stimulation electrodes from ten dystonia patients showing low-frequency direct striato-pallidal pathway activity, indicating a possible pathophysiological role for this pathway. Dystonic symptom severity correlates with frequency-specific coupling pathways. This study presents original mechanistic insight into the pathophysiology of Dystonia by exposing a link between symptom severity and frequency-sensitive coupling.

1) The figures for electrode localization is of bad quality and make anatomical assumptions about the localization of the active contacts impossible. The visualization of the electrode is improper and only schematic and makes also implications on active contacts / directionality / levels where the recordings were applied impossible. This is generally an important point (s. also P4) as also the generalizability of the anatomical specificity is not clear to me. A exact presentation of the each electrode position (not group data) is needed to check the differential localization and the nature of the effects.

We thank the reviewer for the important remark that the localization of individual recording sites were not sufficiently well visualized in the previous version of the manuscript. The correct localization of recording sites within either striatum, GPi or GPe is a central point and priority of this study and we put a maximal effort in selecting only recording sites in which two electrode contacts were localized within the respective structure (either striatum, GPi or GPe) to assure spatial specificity of recorded brain activity by bipolar recordings. In this regard, we have previously validated the spatial precision of our methodological approach built on the DBS imaging toolkit Lead-DBS across raters (Lofredi et al., 2022). Lead-DBS has served as the methodological foundation for many studies from different research labs worldwide, published in Nature Communications (Li et al. 2020; Oswal et al., 2021; Rios et al., 2022; Yin et al., 2023) and other high-impact journals (f.ex. Horn et al., 2017; Mosley et al., 2020; Lofredi et al., 2021). As suggested by the reviewer, we have now replaced the schematic visualization of recording sites and show all recording sites within the target area as reconstructed using the DISTAL atlas (Ewert et al., 2017) for subcortical brain areas (see red dots in Figure 1C, attached below).

The reviewers comments regarding the lack of visualization of “active contacts / directionality / levels” made us realize that we did not make it sufficiently clear in the previous version of the manuscript which electrode design and recording condition was used, as neither directional nor chronic recordings after implantation of the pulse generator are possible with this electrode design and manufacturer. To make this aspect more apparent, we included a schematic of the used electrode design in the revised version of Figure 1A (see below), which shows that the electrode used (*DB-2201-45Bc*, Boston Scientific) is composed of 8 ring contacts without segments. During the short time interval after electrode implantation before lead connection to the pulse generator, we performed simultaneous bipolar recordings from adjacent contact pairs of one electrode, which resulted in 7 parallel recordings from each electrode per hemisphere and 14 parallel recordings per patient as now shown in the revised version of Figure 1B. This electrode type does not allow for directional recordings nor chronic recordings (that latter being available for example with Medtronic stimulators “Percept”). For further analysis, we selected only the bipolar recording pairs where both contacts were localized in the respective structures, i.e. both contacts within the striatum (as shown in purple), GPe (shown in blue) and GPi (shown in green) according to the DBS-electrode localization in standard space. In

one case (patient #6, right hemisphere), one contact of the bipolar recording assigned to the GPi was localized slightly ventral to the GPi in adjacent white matter. Given that white matter does not display oscillatory activity, we decided to still include this case in further analysis.

In addition to the revised version of Figure 1, we added visualizations of individual DBS-electrode localization from all included patients in the Supplementary material. Furthermore, more information describing the electrode design and recording technique has been included to the revised version of the manuscript which now reads as follows:

“The DBS-electrode model used was Vercise™ DBS, DB-2201 (Boston Scientific), with 8 circular contacts spanning over 15.5 mm, see Figure 1 A.”

“DBS-electrodes showed at least two contacts within the striatum, two contacts within the GPe and two contacts within the GPi (see Supp. Fig. 1 and Supp. Table 1), according to segmentation of basal ganglia nuclei in the DISTAL atlas¹⁷, an exemplary case is shown in Fig. 1B. In one case (patient # 6, right hemisphere), one DBS-contact of the GPi-recording was placed slightly ventral to the GPi, in adjacent white matter.”

“Three minutes of rest activity was recorded bipolarly from adjacent contacts in the 1-7 days following surgery while DBS-leads were still externalized. The implanted DBS-electrode (Boston Scientific Vercise™ DBS) is composed of eight circular contacts spanning over 15.5 mm, thereby allowing parallel bipolar recordings of three basal ganglia nuclei (namely the striatum, GPe and GPi). However, due to the electrode design and pulse generator type, no directional steering nor chronic neuronal recordings are available.”

Figure 1 : Methodological set-up and recording sites across basal ganglia nuclei of all patients. (A) In this study, the DBS-electrode Boston Scientific Vercise was implanted in all patients. Note that this DBS-electrode model is composed

of eight circular contacts that span over 15.5 mm. **(B)** Shown is an exemplary recording in the post-operative period. In all patients, we performed bipolar recordings from adjacent contacts resulting in 7 parallel recordings per hemisphere. After DBS-electrode localization, recordings were assigned to either striatum, GPi or GPe if two recording contacts were localized in the respective structure. In the shown example, this would fit to the recording between contact 1 and 2, which are both localized in the GPi (green); contact 5 and 6, both localized within the GPe (blue) and contact 7 and 8, both localized within the striatum (purple). Thus, these recordings were retained for further analysis. In contrast, all other parallel recordings (namely bipolar recordings from contact 2 and 3, contact 3 and 4 as well as contact 6 and 7 as colored in black) are removed from further analysis steps. Thereby, only three parallel recordings per hemisphere (striatum, GPe and GPi) are kept from the original seven. **(C)** Shown are the exact localization within striatum (purple), GPe (blue) and GPi (green) of these retained recording sites from all hemispheres across all patients (red dots). For visualization purposes all recording sites have been flipped to the respective nuclei of the right hemisphere.

Supp. Figure 1: DBS-electrode localization with indication of recording sites for all subjects. Shown are DBS-electrode localization within the basal ganglia nuclei according to the DISTAL atlas for subcortical structures of all included patients. The recording sites included in further analysis are labeled by red dots. For each patient, three recording sites per hemisphere are selected with one lying within the striatum (purple), one within the GPe (blue) and one within the GPi (green). Note that for patient #7 only the right hemisphere is shown, because the left DBS-electrode did not cover the striatum and was thus excluded from further analysis.

2) The general presentation of the results is counterintuitive. Although I appreciate the need for the presentation of single cases for the visualization – the group results are needed to address the main results of the study. The presentation of the results in the Fig 2 (B, C, D) is not comprehensive. The significant contrast are not highlighted and visually relatable.

We thank the reviewer for the comment. We have now clarified the descriptive nature of the figure and main result on the spectral density maps and peak frequency distributions across basal ganglia nuclei. Figure 2 shows for the first time the oscillatory activity patterns across basal ganglia structures in patients with dystonia. This activity pattern is similar across the three nuclei and this is why no significant contrasts are visible in Figure 2. Although no significant differences are shown, we think this figure is helpful, as no parallel recordings from striatum, GPi and GPe in humans have been performed so far. By showing the oscillatory patterns and peak frequencies in the frequency bands of interest, we think that the reader gets a better understanding on activity patterns across the basal ganglia nuclei. As suggested by the reviewer, we rearranged Figure 2 and removed the exemplary case from the revised version of the manuscript to focus on group data. We hope that the presentation of the results regarding oscillatory activity patterns has thereby become more comprehensible and intuitive.

It now reads as follows: *We found a similar pattern of neuronal activity in the three basal ganglia nuclei in all patients. Distinct spectral peaks in the low-frequency range were present in all striatal, all GPi and 17 out of 19 GPe recordings with a mean peak frequencies of about 8.5 Hz across structures and patients (ranging from 3 to 12 Hz). Fig. 2 shows the mean peak frequency for striatum: 8.5 ± 2.5 Hz; GPi: 8.4 ± 2.9 Hz and GPe: 8.7 ± 2.1 Hz across the 19 hemispheres.*

Please find the revised version of the Figure below:

Figure 2: Spectral patterns are similar across basal ganglia nuclei in dystonia patients. (A) When averaged across hemispheres of all subjects, similar spectral patterns are observed with spectral peaks in the low-frequency (upper row) and beta band (lower row) in the striatum (STR, left column), GPi (mid column) and GPe (right column). Averaged power spectra across hemispheres and patients are colored (STR=purple, GPi=green, GPe=blue), while individual power spectra of each hemisphere are plotted in grey. For visualization purposes, power spectra are flattened and aligned to the individual peak frequency. No significant difference in power spectral density at peak frequency is seen across basal ganglia nuclei (not shown). **(B)** Similarly, peak frequencies in the respective frequency bands (low frequency and beta band) did not significantly differ across basal ganglia nuclei and were distributed across the low frequency (upper row) and beta band (lower row). In the box plots, central marks indicate the median and edges the 25th and 75th percentiles of the distribution.

3) One of the main concerns is a possible acute nature of the effects. With the insertion of the electrodes a microlesion effect occurs and leads to an acute reorganization of the brain networks. The activity in the internal pallidum and also the synchronization between striatum and internal pallidum could be related to symptom improvement or the microlesional effect. The authors should exclude this mechanistic explanation with data from chronically implanted patients. A further analyses that could bring an improved causality proof of the results would be the analysis of responders' vs non-responders (i.e. also from the postoperative acutely implanted patients with the micro-lesional effect).

We agree with the reviewer that as recordings were performed in the days following DBS electrode implantation, we cannot fully rule out that a potential microlesion effect could have an influence on recordings. However, in our opinion, the resulting effect would be expected to bias against our findings. Most importantly, the potential neurophysiological effect of a brain lesion, even if in the micro-scale, is expected to be a loss of neural activity due to edema and tissue damage.

Only a handful of studies have specifically studied the microlesion effect in any DBS indication and no study is available for dystonia. For Parkinson's disease, a paper followed three patients over days and observed a significant suppression in LFP amplitude on the first day after implantation in 2/4 reported subjects followed by a recovery of LFP activity (Brinda et al., 2021). A similar trend was reported in a very recent paper (Peng et al., 2024) and a poster abstract of the MDS conference (<https://www.mdsabstracts.org/abstract/postoperative-change-of-local-field-potentials-of-subthalamic-nucleus-during-and-after-period-of-microlesion-effect-of-implantation-of-deep-brain-stimulation-lead/>), which is consistent with our own preliminary findings in PD.

Notably, these studies suggest that if any effect, a suppression of LFP activity may result from the microlesion effect. However, it is important to point out that all these studies were derived from patients with a different disease (Parkinson's disease), recorded with a different electrode (none used Boston Scientific Vercise) and in a different brain area (all STN). We understand that, perhaps if not through direct changes at the tissue interface, a potential change in the LFP patterns could result indirectly from clinical improvement. This is commonly observed in STN-DBS for Parkinson's disease, and some authors have argued that if pathological LFP activity has causal implications for clinical symptoms, an interaction of microlesion, suppression of oscillations and clinical phenomenology is to be expected. Translating this into our study on dystonia, we would indeed expect decreased oscillatory activity in the low frequency range in patients who exhibit less symptoms post-surgery. However, it is important to note that the time-course of symptom alleviation of dystonia is remarkably different, and it often takes months after DBS surgery for patients to improve. Most importantly, this would be expected to be accompanied by less oscillatory synchrony. In summary, we acknowledge a potential microlesion effect, but hope to convince the reviewer that a spectrally specific bias towards the results we report is not plausibly attributable to microlesion effects.

Nevertheless, we strongly agree with the reviewer that following these patients over months with chronic recordings from striatum, GPi and GPe would provide further important insights. Unfortunately, the technology required for that is currently not available. The only commercially available DBS-lead with 8 circular contacts that span over 15.5 mm and therefore allows parallel recordings from the striatum, GPe and GPi is the "Vercise" electrode by Boston Scientific. With these electrodes, recordings cannot be repeated after the implantation of the pulse generator (denominated "chronic recordings" by the reviewer), as the required technology is not available. We have extensive experience with sensing enabled devices, such as the first generation Medtronic PC+S and the Medtronic Percept, but these are incompatible with the electrode model that provides access to striatal activity for pallidal trajectories reported here, because the electrode models do not have sufficient span. We are hopeful that revealing for the first time the pathophysiological relevance of these parallel recordings from several basal ganglia nuclei and the potential

role for adaptive DBS algorithms might change this situation and motivate manufacturers such as Medtronic to additionally provide new DBS-electrode designs to their sensing-enabled pulse generators.

In conclusion, as we are the first to have recorded striato-pallidal coupling patterns, we cannot prove their robustness in chronic recordings. This is why we have decided to emphasize this methodological constraint and highlight the need for further development of chronic circuit recordings in the revised version of the manuscript, which now reads as follows:

“Notably, our results have been obtained in the acute perioperative period, where surgical intervention and patient state may have affected neural recordings, e.g. through the so-called microlesion effect. With the rise of new sensing enabled devices, it will be important to extend these findings to dynamic network changes with chronic DBS, alongside the symptom alleviation which can take up to months after implantation.”

“These findings from the acute, post-operative period may inspire the development of new neurotechnological devices that allow for chronic recordings of several basal ganglia nuclei in order to develop a circuit-based approach to neuromodulation therapies and incorporate pathophysiological network signatures as feedback signals for novel and demand-adapted treatment strategies.”

4) A further major drawback is the regional / anatomical specificity of the effects. The synchronization could be related to the influence of a distant pacemaker (i.e. sensory cortex) that sends then a different input to striatum / GPi/ GPe structures.

We thank the reviewer for pointing us to this conceptual discussion. We agree with the reviewer that it is unlikely that the basal ganglia stand isolated in the generation of pathophysiological brain circuit changes and we agree that extending the recording protocols to include more brain regions will provide further information on disease mechanisms. Having that said, in a previous study (Neumann et al., 2015), we combined the pallidal recordings with whole-head magnetoencephalography (MEG) and found no clear link between pathological low frequency oscillations and sensory or motor cortex, but observed connectivity with the cerebellum that showed a significant correlation with dystonic features. This has in part inspired the extension of the subcortical circuit discovery reported here, as the cerebellum has no direct input to the GPi. Instead, like cortex, it communicates with the basal ganglia via the striatum, the main input structure of the basal ganglia. Taking this into account, we reasoned that regardless of the potential additional sources of external input, be it from cortex, thalamus or other subcortical nuclei, the major pathways will all integrate at the level of striatum. Moreover, we took inspiration from Kaji, Bhatia & Graybiel (2018) and nuclear imaging (Simonyan et al., 2017) that proposed a strong a priori hypothesis on the role of the direct pathway between striatum and internal pallidum in the pathophysiology of dystonia (Kaji et al., 2018; Simonyan et al., 2017). But to this date, neurophysiological evidence for this hypothesis in humans was lacking, which can be attributed to the notorious difficulty of recording neurophysiology from multiple deep structures in parallel. Being able to address this knowledge gap in the present study has become possible through the use of a specific electrode type and precision neuroimaging. After identifying this new opportunity, we wanted to tackle to what degree direct and indirect pathway structures contribute to the observed pathophysiology. We think that our finding of a correlation between striato-pallidal coupling strength and symptom severity that is specific for the internal and not the external pallidum is an important contribution. The absence of this correlation for connectivity with the GPe indeed has a major implication for the reviewers concern. Given that cortical areas have low synaptic specificity for indirect vs. direct pathway medium spiny neurons at the level of striatum, it is unlikely that the critical pathophysiological change is generated in cortex, because if that was to be the case, we would either expect correlations across all basal ganglia structures or no correlations at all. Thus, while we agree that cortical or cerebellar input may be necessary to drive striatal activity, our findings hint towards specific changes in oscillatory synchrony that occur within the basal ganglia.

Nevertheless, we still acknowledge that our study may not suffice to rule out potential cortical sources and thus we decided to further discuss this possibility in the revised version of the discussion section, which now reads as follow:

“Dystonia is increasingly understood as a circuit disorder and many brain areas beyond the basal ganglia have been implicated in its’ pathophysiology, such as (pre-)motor and sensory cortices²⁴ as well as the cerebellum^{25,26}. Thus, it is unclear, whether the observed abnormal neuronal pattern are spreading from these brain areas via the extensive connectivity of striatum or are of local origin within the basal ganglia. Here, we report oscillatory activity patterns that are thought to reflect afferent synaptic voltage fluctuations¹¹. Thus, if pathological low-frequency signaling would arise upstream to the basal ganglia, oscillatory power in the striatum, as input nucleus of basal ganglia, should already show a link to dystonic symptom severity. The observation that the link between low-frequency power and dystonic symptoms is lacking in the striatum and specifically found in the GPi and for connectivity between direct striatopallidal pathway but not indirect striatum-GPe or GPe-GPi, may thus be interpreted as hint toward a basal ganglia source of this pathological phenomenon. Still, follow-up studies with parallel electrocorticography (ECoG) or whole-head M/EEG recordings and basal ganglia activity could reveal additional mechanisms on the interplay between pathophysiological basal ganglia activity and other hubs of the sensorimotor circuit.”

5) Further I am relatively concerned about the novelty of the work. A rather vast literature exist on the role of low-oscillation for the generation of dystonic symptoms and their relation to therapeutic responses to DBS (i.e. see <https://pubmed.ncbi.nlm.nih.gov/34328685/> <https://pubmed.ncbi.nlm.nih.gov/37268239/> <https://pubmed.ncbi.nlm.nih.gov/30440096/>

A more in-depth discussion of the novelty and pathophysiological clinical implications is needed.

We hope that we could address this in part in the responses to 1-4, but will further extend here. Our work is based on the previous finding of low frequency activity in the internal pallidum being associated with dystonic symptom severity. Given that dystonia is recognized as network disorder and the striatum holds a crucial role in the network that is supposed to underly the pathophysiology of dystonia (see Kaji et al., 2017; Corp et al., 2019), a basal ganglia circuit characterization that specifically includes the striatum is a fundamental next step for both a better understanding of the underlying pathophysiology as well as an identification of alternative target structures for future DBS-algorithms. It is exactly this step that we are taking with this study, signifying both its novelty and its potential clinical implication.

This study is the first to perform a circuit investigation of several basal ganglia nuclei, instead of recording from a single basal ganglia nucleus, such as the internal pallidum or in rare case the subthalamic nucleus of dystonia patients. Indeed, in the vast majority of previous studies, intracerebral recordings were restricted to the internal pallidum, the main DBS-target nuclei for dystonia, as also shown by the references cited by the reviewer. In contrast, activity patterns in the striatum of dystonia patients were investigated only in two studies (Singh et al., 2016; Valsky et al., 2020) and both focused on firing rates via multi-unit recordings. Neither of these two studies had the technical possibility to record other basal ganglia nuclei in parallel for a circuit analysis, nor did the authors assess the link between striatal activity to dystonic symptom severity. In contrast, this study is not only the first to investigate the link between dystonic symptom severity and oscillatory recordings from the striatum – which by itself would already be a novelty compared to previous studies -, it also performs the first basal ganglia circuit characterization by recording from several basal ganglia nuclei in parallel. This approach and access to several basal ganglia nuclei, specifically including the striatum, has not been possible before as it is tightly linked to neurotechnological developments such as the specific DBS-electrode design used in this study.

Dystonia is considered a network disorder, where an imbalance towards the direct pathway within the basal ganglia is presumed (Kaji et al., 2017; Simonyan et al., 2017). However, so far there has been no neurophysiological evidence for this hypothesis. By performing basal ganglia circuit recordings in this study, we can show for the first time that there is an association between circuit activity between striatum and internal pallidum that we don't find for the circuit between striatum and external pallidum. This indicates that indeed the connection between striatum and internal pallidum, known as the direct pathway, is involved in the pathophysiology of dystonia. Based on this, our results will also help to guide developments to improve treatment options for dystonia by stronger consideration of striatal and striato-pallidal signaling for adaptive DBS-algorithms in dystonia.

Current treatment strategies such as pallidal or subthalamic DBS show a large variance in outcome across dystonia patients and it can be challenging to reach optimal effects due to delayed DBS-effects and the large DBS parameter space. Thus, identifying neurophysiological markers of symptom severity that can be recorded chronically and guide automatized adaptation of DBS-settings will be a hallmark in future DBS-paradigms. In this study, we report a new, promising neurophysiological marker that is striato-pallidal low frequency coupling.

Ultimately, we sympathize with the reviewers comment 4 questioning the potential network source of pathophysiological activity and we believe that novel studies that directly address these questions through exceptionally ambitious multisite recording approaches are needed to gain a better mechanistic understanding. We believe that our present study demonstrates the feasibility of such ambitious approaches and hope to inspire future studies and devices that further extend the network coverage in space, through multielectrode approaches, or time, through availability of chronic recordings.

We hope that we could convince the reviewer of the novelty as well as the implications for pathophysiological understanding and development of new treatment avenues for dystonia. We further emphasized these aspects in the discussion section which reads as follows in the revised version of the manuscript:

“These results integrate into a large body of evidence suggesting that altered patterns of synchronization in neuronal assemblies as recorded by oscillatory activity may underlie pro- and anti-kinetic signaling within basal ganglia pathways^{16,17}. Locally, exaggerated synchronization in beta oscillations has been linked to the occurrence of hypokinetic symptoms^{6,7,18}, while low-frequency and gamma oscillations have been associated with voluntary and involuntary movements across movement disorders^{8,10,19–23} and thus been qualified as pro-kinetic. Due to technical constraints, previous studies were restricted to recording sites within single nuclei of the basal ganglia, mostly DBS-target structures such as the GPi or STN. Importantly, our study is the first to connect these local observations to coupling patterns across the basal ganglia, as parallel recordings of several basal ganglia nuclei, most importantly including the striatum, have been lacking in humans.”

“These findings from the acute, post-operative period may inspire the development of new neurotechnological devices that allow for chronic recordings of several basal ganglia nuclei in order to develop a circuit-based approach to neuromodulation therapies and incorporate pathophysiological network signatures as feedback signals for novel and demand-adapted treatment strategies.”

6) The use / relevance of the connectomic analyses are not clear to me. The work makes assumptions to the interconnectivity to the addressed network nodes but no intra-subject demonstration of the network effects with the applied methods is possible. The use of the “generalistic” vs individual connectome is a major drawback. Furthermore this a group of dystonia patients, where no work was done (to my best knowledge) on the comparability of the rs-fMRI between the group and the interapicability.

In this study, we performed exact DBS-electrode localization in normative MNI-space using the Lead-DBS software. Thereby, we made sure that all recording sites were lying within the structures that we assigned them to (striatum, GPe and GPi, respectively). It is on these recordings that rely the main findings of this work after careful anatomical validation. But we were not satisfied with a mere anatomical “nuclei” definition, as we know that the basal ganglia follow a functional gradient and we were limited to recording sites along the implant trajectory. Therefore, we wanted to make sure that the recording source is functionally relevant by verifying that the striatal recording location is not primarily connected to limbic domains, while the pallidal recording location lies in the sensorimotor region. To satisfy our ambition, we consulted the connectome as a “normative wiring diagram” of the human brain, to ensure that each of the recording sites is of motor origin. We agree with the reviewer that individual data could have been additionally interesting, but 1) acquisition of fMRI-data is not possible in patients with externalized DBS-electrodes and 2) normative neuroimaging shows an improved signal-to-noise ratio when compared to individually acquired fMRI-data. Specifically for dystonia, normative connectivity analyses have been recently validated to explain DBS-effects in both cervical and generalized dystonia (Horn et al., 2022). This provides evidence that normative connectomes indeed reflect the connectivity profile in dystonia patients. However, as the connectivity analysis in this study is not part of the main results, we decided to shift the Figure in which the connectivity profile of the recording sites is shown to the Supplementary Material and further emphasize in the revised version of the manuscript that the normative connectivity profile is just an additional piece of evidence indicating the association with the motor network of our basal ganglia recording sites. Please find below the new version of a Supp. Figure 2 as well as the changes in the revised version of the manuscript:

“The trajectory of DBS-electrode insertion led to similar, yet slightly more anterior recording sites in DBS-electrode contacts localized in the posteroventral striatum. To assure their localization within the motor circuit, we conducted a connectomic analysis of the patient individual recording sites within the respective basal ganglia nucleus, shown in Supp. Figure 2. Whole-brain connectivity in an openly available group connectome was estimated by using a 2 mm seed roughly reflective of the neural field recorded from DBS contacts (for more details regarding this approach see Supp. Materials). This analysis provides additional evidence for the recording sites being part of the sensorimotor circuit within the basal ganglia, but does not enable the investigation of symptom-specific changes in neuroimaging-based connectivity.”

fMRI-connectivity profiles of recording sites in

Supp. Fig. 2: Normative connectivity profiles of recording sites within the basal ganglia. The averaged functional connectivity of recording sites within each basal ganglia nucleus (left column: striatal recording sites; mid-column: GPi recording sites; right column: GPe recording sites) corresponds to the connectivity profile of their sensorimotor portion. For example, striatal recording sites show positive coupling to motor cortex, cerebellum and supplementary motor areas (yellow-red) and negative coupling to sensory cortices (blue-green). Note that these analyses have been performed within normative connectomes.

7) The authors should comment on the concatenated inclusion of cervical and generalized dystonia patients and the relevance of the results. Do the authors see any differences among these groups?

Dystonia is a relatively rare movement disorder with currently unknown etiology. Recent studies suggest that across dystonic phenotypes, the disturbed network parts may increase along with the affected body parts. This would explain why effective DBS in generalized dystonia can interfere with fiber tracts that are more widely distributed across the homunculus while in cervical DBS, the connected fibers most decisive of treatment outcome are those linked to the neck representation in sensorimotor cortices (Horn et al., PNAS; 2022). In this study, we show that LF-power in the GPi as well as LF-coherence between GPi and striatum correlates with symptom severity as measured by the BFMDRS. We show that across dystonia subtypes, this correlation holds true. Thus, there is no indication that LF-activity is a phenotype specific marker, but rather a symptom-specific marker that scales with the extent and severity of dystonia observed, independently of the body part where it occurs. This matches findings of previous studies that investigated oscillatory patterns in single basal ganglia nuclei and concatenated different dystonia phenotypes (Scheller et al., 2019; Neumann et al., 2017; Lofredi et al., 2019; Zhang et al., 2022). However, following the reviewers suggestions we performed a sub analysis of spectral features such as local power, peak frequency and coherence in the LF-band. There was no significant difference in any of those features when comparing patients with cervical dystonia (n=6 patients, n=11 hemispheres) to those with segmental or generalized dystonia (n=4 patients, n=8 hemispheres), as shown in the table below. We added this information to the revised version of the manuscript, which now reads as follows:

“There was no significant difference in spectral power or peak frequency in the LF-band, when comparing cervical (6 patients, 11 hemispheres) with segmental or generalized dystonia (4 patients, 8 hemispheres), see Supp. Table 2.”

“Imaginary part of coherence revealed significant coupling over the averaged low-frequency band across all structures when compared to shuffled data (see Fig. 3; STR-GPi: iCOH=0.07±0.02, shuffled=0.02±0.006, P=0.002; STR-GPe: iCOH=0.066±0.027, shuffled=0.03±0.0096, P=0.003; GPi-GPe: iCOH=0.067±0.034, shuffled=0.02±0.009, P=0.003), which showed no significant difference between cervical (6 patients) and segmental or generalized dystonia (4 patients), see Supp. Table 2.”

BG nucleus	LF-Power			LF-Peak Frequency			Connectivity between BG nuclei	LF-iCOH		
	Cervical	Seg/Generalized	P-Val	Cervical	Seg/Gen	P-Val		Cervical	Seg/Gen	P-Val
Striatum	5.8±1.8	5.8±1.8	0.9	8.1±2.1	9.0±3.3	0.4	STR-GPi	0.09±0.05	0.1±0.006	0.2
GPi	5.2±1.2	5.9±0.9	0.2	8.3±2.5	8.9±3.8	0.6	STR-GPe	0.13±0.09	0.12±0.07	0.6
GPe	5.3±1.0	6.1±2.2	0.3	8.9±1.4	8.2±3.4	0.5	GPi-GPe	0.09±0.045	0.09±0.038	0.9

Supp. Table 2: Subgroup-Analysis of neurophysiological features between patients with cervical or segmental/generalized dystonia. Shown are mean ± standard deviation. Abbreviations: BG nucleus: Basal ganglia nucleus; GPi: Globus pallidus internus; GPe: Globus pallidus externus; LF: Low frequency (3-12 Hz); iCOH: imaginary part of coherence; Cervical: Cervical dystonia; Seg/Gen: Segmental or generalized dystonia; P-Val: p-Value for comparison between cervical and segmental/generalized dystonia group.

8) The use of abbreviations is not conclusive (STR is not introduced)

We thank the reviewer for having noticed the missing introduction of “STR” as abbreviation for striatum. We have included this in the revised version of the manuscript, which now reads as follows:

“Within the basal ganglia, PET studies demonstrated a specific increase of striatal D1- and a decrease of D2-receptors that was attributed to a potential hyperactivity of the direct pathway between the striatum (STR) and the globus pallidus internus (GPI)^{4,5}.”

9) „Imaginary“ part of coherence – is misleading

We agree with the reviewer that this wording can be puzzling. To understand our choice to include this wording, we would like to provide a bit more background of its origin: It is derived from “coherence”, which measures the correlation between two signals at a given frequency. It is a complex quantity, which means it can be broken down into real and imaginary parts, and from these, the absolute coherence can be derived. The imaginary part of coherence represents the “out-of-phase” component of the correlation between two signals. It indicates how much of the signal correlation is due to components that do not have zero phase lag. Given the proximity of recording sites, neuronal coupling could be contaminated by volume conduction (i.e. resulting from an external oscillator, such as cortex or attributable to only the largest structure in proximity to the contacts, e.g. striatum), which would result in zero-phase lag coupling. Thus, the “imaginary part of coherence” component can reveal important insights about the phase relationship between the signals and the non-zero imaginary part suggests that there is a significant phase difference between the signals at that frequency. This ensures that connectivity results from oscillatory synchrony between multiple oscillators, as required to be physiologically meaningful for striatopallidal communication. Some more strict authors have separated the complex measure from the absolute and imaginary derivatives by naming the complex measure “coherency”, the absolute derivative “coherence” and the non-zero phase lag derivative “imaginary part of coherency” as introduced in the corresponding methodological paper by Nolte et al., 2004. But this has not been followed consistently in the literature and may spark even more confusion. Thus, we have deliberately chosen the term “imaginary part of coherence” to describe the frequency domain measure of linear phase and amplitude relationships between signals, while removing instantaneous interactions, as it was most commonly done in publications assessing functional coupling across different sources of neuronal population activity. We hope that the reviewer thus understands that we refrain from changing the wording with respect to the results referring to this methodological approach. However, we have clarified this further in text:

“Coherence measures the correlation between two signals at a given frequency, a complex quantity, which can be broken down into real and imaginary parts. The latter part, reflects connectivity without zero phase lag, indicative of true communication of distinct oscillators, without contamination from volume conduction. For our analysis we thus relied on this more conservative and robust connectivity measure. Given that both complex and absolute measures are commonly called “coherence”, we follow the previously suggested terminology and refer to “imaginary part of coherence” in the present text.”

10) The relevance of the MEG recordings / analyses is not clear to me and should be excluded (or better elaborated)

As suggested by the reviewer, we excluded the information that parallel MEG recordings were performed in two participants from the revised version of the manuscript, as the parallel MEG-recordings were not analyzed in the current study.

Reviewer #2 (Remarks to the Author):

This paper looked into the neurophysiological coupling among the striatum, the external and the internal pallidum with unique simultaneously recorded local field potentials. The research provides insights into the functional coupling within direct striato-pallidal pathway in dystonia. The study confirmed previous

findings that the low frequency oscillations in GPI are correlated to the dystonic symptom severity. More interestingly, dystonic symptoms correlate specifically with low-frequency coupling between striatum and internal pallidum. The local field potentials were recorded from adjacent contacts and there may be volume conduction within such local region. The imaginary part of coherence was used to reduce the influence of volume conduction, but it may cause false positive coherency (Ghost interactions in MEG/EEG source space: A note of caution on inter-areal coupling measures. DOI10.1016/j.neuroimage.2018.02.032). It would be essential to be cautious to such measures and exclude the spurious interaction. It might be useful to use another measure, for instance, weighted phase lag index, to confirm the findings.

We thank the reviewer for their positive evaluation of our study. We followed their suggestion and calculated the weighted phase-lag-index (wPLI) as measure of neuronal coupling in addition to imaginary part of coherence. Thereby, we made sure that the reported results of basal ganglia coupling are not linked to a false positive measure of neuronal coupling derived by a single methodological approach such as coherence. We are happy to report that we could fully reproduce the low frequency coupling with the weighted phase lag index measure. Indeed, this purely phase based method even revealed phase coupling in the beta range, which matches the observed beta peaks across all basal ganglia nuclei, but was not detected by imaginary part of coherence. Perhaps this could be explained by the dissociation of amplitude and phase for wPLI. In contrast, coherence takes not only phase but also amplitude into account. Given the more intuitive hypothesis that previously reported local amplitude changes also reflect in interregional phase-amplitude connectivity, we decided to stick with imaginary part of coherence for the main analysis but have added a supplementary figure visualizing the spectral distribution of the weighted phase-lag-index of original when compared to shuffled data in the revised version of our manuscript. We included the additional findings in our methods and result section, which now reads as follows:

“In addition, we calculated the weighted phase-lag-index as a supplementary measure for neuronal coupling, to make sure that the reported results did not rely on false positive coherency (Palva et al., 2018).”

“These findings could be reproduced by using the weighted phase-lag-index (wPLI) that showed significantly higher coupling in the low frequency band when compared to shuffled data (see Supp.Fig.3; STR-GPi: $wPLI=0.14\pm0.05$, shuffled= -0.001 ± 0.01 , $P=0.002$; STR-GPe: $wPLI=0.16\pm0.13$, shuffled= 0.0049 ± 0.01 , $P=0.004$; GPI-GPe: $wPLI=0.14\pm0.1$, shuffled= 0.00001 ± 0.01 , $P=0.002$).”

Supplementary Figure 3: Neuronal coupling across basal ganglia nuclei as measured by weighted phase-lag-index (wPLI). (A) Shown are averaged spectra of wPLI between striatum and GPe (purple), striatum and GPi (blue), GPe and GPi (green) and averaged shuffled wPLI (grey) across hemispheres. (B) All recorded basal ganglia structures are

functionally coupled in the low-frequency band as shown by significantly higher wPLI values when compared to shuffled data. ** $p < 0.01$. Black dots indicate mean value per hemisphere. In box plots, central marks indicate the median and edges the 25th and 75th percentiles of the distribution.

Reviewer #3 (Remarks to the Author):

The authors investigated how indirect and direct pathways of the basal ganglia circuits are involved in the pathophysiology of dystonia. This is the first study in humans analyzing parallel measurements from the Striatum, the GPe, and the GPi. The authors examined 20 patients and analyzed 19 hemispheres. With Boston Vercise standard leads, they placed two ring contacts with one insertion to the three targets. They measured signals from the contact pairs in a bipolar configuration. Functional connectivity with motor cortex areas and the cerebellum confirmed the correct electrode locations. The working group used an innovative method to map the direct and indirect basal ganglia network activity. The results are consistent with previous literature data.

The results:

They found a 3-12Hz activity and a less frequent 13-35Hz activity in all three targets without a difference in the peak frequency or power across the structures. They revealed low-frequency, bidirectional information flow between the Striatum and the GPi, in which the Striatum dominated in more cases. They found a significant positive correlation between the low-frequency power in the GPi, the low-frequency coupling of the Striatum and GPi, and the severity of dystonia.

This explorative study has an appropriate number of subjects. The study planning and the method used are exact. The statistical analysis is well performed. The presentation of the results is logically structured, and the Results paragraph is easy to follow. The whole article is enjoyable to read; the essential results are adequately emphasized in the text.

Comments:

1. In the legend of Fig 3A, please define the light-color intervals.
2. The last two sentences of the Fig 3 legend should relocate to the legend of Fig 3B.

In summary, this is an excellent and innovative work. I recommend publishing the manuscript in the journal.

Yours sincerely,

Gertrud Tamas

We thank Prof. Tamas for these positive comments and have adjusted the Figure legend according to her suggestion. The Figure legend of Fig. 3 now reads as follows in the revised version of the manuscript:

Figure 3: Connectivity patterns across basal ganglia structures and their correlation to symptom severity. (A) Shown are averaged coherence spectra between striatum and GPe (purple), striatum and GPe (blue), GPe and GPi (green) and averaged shuffled coherence (grey) across hemispheres. Shaded areas indicated standard deviation of the mean. (B) All recorded basal ganglia structures are functionally coupled in the low-frequency band as shown by significantly higher coherence values when compared to shuffled data. Black dots indicate mean value per hemisphere. In box plots, central marks indicate the median and edges the 25th and 75th percentiles of the distribution. (C) Shown are averaged spectra of granger causality with GPi (blue) or striatum (purple) as source. (D) When each frequency bin is considered separately, the majority of the low-frequency band is led by the striatum in 63% and by the GPi in 37% of cases. (E) Dystonic symptom severity correlates both with averaged low-frequency power in the GPi and (D) the coupling strength of low-frequency activity between the striatum and the GPi. ** $p < 0.01$.

References

- Brinda, A. K. et al. Longitudinal analysis of local field potentials recorded from directional deep brain stimulation lead implants in the subthalamic nucleus. *J. Neural Eng.* **18**, (2021).
- Corp, D. T. et al. Network localization of cervical dystonia based on causal brain lesions. *Brain* **142**, 1660–1674 (2019).
- Ewert, S. et al. Toward defining deep brain stimulation targets in MNI space: A subcortical atlas based on multimodal MRI, histology and structural connectivity. *Neuroimage* **170**, 271–282 (2018).
- Horn, A. et al. Connectivity Predicts deep brain stimulation outcome in Parkinson disease. *Ann. Neurol.* **82**, 67–78 (2017).
- Horn, A. et al. Optimal deep brain stimulation sites and networks for cervical vs. generalized dystonia. *Proc. Natl. Acad. Sci. U. S. A.* **119**, e2114985119 (2022).
- Kaji, R., Bhatia, K. & Graybiel, A. M. Pathogenesis of dystonia: is it of cerebellar or basal ganglia origin? *J. Neurol. Neurosurg. Psychiatry* **89**, 488–492 (2018).
- Li, N. et al. A unified connectomic target for deep brain stimulation in obsessive-compulsive disorder. *Nat. Commun.* **11**, 3364 (2020).

- Lofredi, R. et al. Pallidal beta bursts in Parkinson's disease and dystonia. *Mov. Disord.* **34**, (2019).
- Lofredi, R. et al. Interrater reliability of deep brain stimulation electrode localizations. *Neuroimage* **262**, 119552 (2022).
- Lofredi, R. et al. Subthalamic stimulation impairs stopping of ongoing movements. *Brain* **144**, 44–52 (2021).
- Mosley, P. E. et al. The structural connectivity of subthalamic deep brain stimulation correlates with impulsivity in Parkinson's disease. *Brain* **143**, 2235–2254 (2020).
- Neumann, W. et al. Cortico-pallidal oscillatory connectivity in patients with dystonia. 1894–1906 (2015) doi:10.1093/brain/awv109.
- Neumann, W.-J. et al. A localized pallidal physiomaer in cervical dystonia. *Ann. Neurol.* **82**, 912–924 (2017).
- Oswal, A. et al. Neural signatures of hyperdirect pathway activity in Parkinson's disease. *Nat. Commun.* **12**, 5185 (2021).
- Palva, J. M. et al. Ghost interactions in MEG/EEG source space: A note of caution on inter-areal coupling measures. *Neuroimage* **173**, 632–643 (2018).
- Ríos, A. S. et al. Optimal deep brain stimulation sites and networks for stimulation of the fornix in Alzheimer's disease. *Nat. Commun.* **13**, 7707 (2022).
- Scheller, U. et al. Pallidal low-frequency activity in dystonia after cessation of long-term deep brain stimulation. *Mov. Disord.* **34**, 1734–1739 (2019).
- Singh, A. et al. Human striatal recordings reveal abnormal discharge of projection neurons in Parkinson's disease. *Proc. Natl. Acad. Sci. U. S. A.* **113**, 9629–9634 (2016).
- Valsky, D. et al. What is the true discharge rate and pattern of the striatal projection neurons in Parkinson's disease and Dystonia? *Elife* **9**, e57445 (2020).
- Yin, Z. et al. Pathological pallidal beta activity in Parkinson's disease is sustained during sleep and associated with sleep disturbance. *Nat. Commun.* **14**, 5434 (2023).
- Zhang, S. et al. Clinical features and power spectral entropy of electroencephalography in Wilson's disease with dystonia. *Brain Behav.* **12**, e2791 (2022).

REVIEWER COMMENTS

Reviewer #1 (Remarks to the Author):

Many thanks for the in-depth revisions. However I further see the drawbacks (novelty, topographic specificity of the results) of this work.

Specifically

- Visualisation of the results / recording electrodes. The presented 2D planes with clear anatomical borders of the segmented structures are helpful, however a further visualisation of the medial-lateral plane and ventral-dorsal is also needed.
- The visualisation of the statistics for the Figure 2 is missing
- Please present the GPe, Striatum vs BFMDRS, clinical scores. That would underpin the topological specificity of the results

Reviewer #2 (Remarks to the Author):

The paper presented exploratory investigations into the direct and indirect pathways of the basal ganglia circuits involved in the pathophysiology of dystonia. It is unique and intriguing although there might be some uncertainties to be further clarified in the future.

One suggestion is to provide details on the DBS targeting and implantation procedures in supplement. The location of all electrodes looks too good to be true. Additional information on the procedure and original MRI scannings would further consolidate the results as LeadDBS doesn't always work well.

Reviewer #3 (Remarks to the Author):

The authors have corrected what I requested, and I have no other comments. I repeat my opinion that I recommend publishing the article in the journal.

Reviewer #1:

1) Many thanks for the in-depth revisions. However I further see the drawbacks (novelty, topographic specificity of the results) of this work. Specifically

-Visualisation of the results / recording electrodes. The presented 2D planes with clear anatomical borders of the segmented structures are helpful, however a further visualisation of the medial-lateral plane and ventral- dorsal is also needed.

According to the reviewers suggestion, we extended our supplementary figure that shows the individual DBS electrode localizations (Supp. Fig. 1) in the revised version of the manuscript by adding the dorso-ventral and medio-lateral view as shown below. We reference this supplementary figure in the revised version of the manuscript as follows:

“DBS-electrodes showed at least two contacts within the striatum, two contacts within the GPe and two contacts within the GPi (see Supp. Fig. 1 and Supp. Table 1), according to segmentation of basal ganglia nuclei in the DISTAL atlas³², an exemplary case is shown in Fig. 1B.”

Supp. Figure 1: Different views of DBS-electrode localization with indication of recording sites for all subjects. Shown is the localization of DBS-electrodes within the basal ganglia nuclei according to the DISTAL atlas for subcortical structures of all included patients from the posterior view (column 1), dorso-ventral view (column 2) as well as the medio-lateral view of the left (column 3) and the right hemisphere (column 4). The recording sites included in further analysis are labeled by red dots. For each patient, three recording sites per hemisphere are selected with one lying within the striatum (purple), one within the GPe (blue) and one within the GPi (green). Note that for patient #7 only the right hemisphere is shown, because the left DBS-electrode did not cover the striatum and was thus excluded from further analysis.

2) The visualization of the statistics for the Figure 2 is missing

As shown below, we added the information that peak frequencies show no statistical difference across basal ganglia nuclei to the revised version of Figure 2:

Figure 2: Spectral patterns are similar across basal ganglia nuclei of dystonia patients. (A) When averaged across hemispheres of all subjects, similar spectral patterns are observed with spectral peaks in the low-frequency (upper row) and beta band (lower row) in the striatum (STR, left column), GPi (mid column) and GPe (right column). Averaged power spectra across hemispheres and patients are colored (STR=purple, GPi=green, GPe=blue), while individual power spectra of each hemisphere are plotted in grey. For visualization purposes, power spectra are flattened and aligned to the individual peak frequency. No significant difference in power spectral density at peak frequency is seen across basal ganglia nuclei (not shown). **(B)** Similarly, peak frequencies in the respective frequency bands (low frequency and beta band) did not significantly differ across basal ganglia nuclei and were distributed across the low frequency (upper row) and beta band (lower row), n.s. = not significant. In box plots, central marks indicate the median and edges the 25th and 75th percentiles of the distribution.

3) Please present the GPe, Striatum vs BFMDRS, clinical scores. That would underpin the topological specificity of the results

In order to further demonstrate the topological specificity of the correlation between the BFMDRS and low frequency coherence between the GPi and striatum, we extended Figure 3 in the revised version of the manuscript by adding the non-significant correlation plots between the BFMDRS and striatal LF power (Fig 3 E) and the “indirect pathway” activity of LF coherence between striatum and GPe (Fig. 3F). Thereby, Figure 3E now visualizes the contrast between the significant correlation between the BFMDRS and LF power for the main output structure of the basal ganglia which is the GPi ($R=0.88$, $P=0.001$) and the non-significant correlation between the BFMDRS and LF power for the main input structure of the basal ganglia, which is the striatum ($R=0.4$, $P=0.13$). In addition, Fig.3F now shows the distinction of the significant correlation of LF coherence between STR and GPi (as potential correlate of “direct pathway” activity) with the BFMDRS ($R=0.74$, $P=0.0093$) and the non-significant correlation of LF coherence between STR and GPe (as potential

correlate of the “indirect pathway”) with the BFMDRS ($R=-0.029$, $P=0.462$). The findings highlight that it is primarily i) the LF power in GPi (as BG output nucleus) but not striatum (related to input signal from cortex) and ii) LF coupling between striatum and GPi – as potential read-out of “direct pathway” activity – but not the coupling between striatum and GPe – as potential read-out of “indirect pathway” activity – that correlates with dystonic symptoms. The corresponding paragraph in the revised version of the manuscript now reads as follows:

“Correlation of low frequency activity and symptom severity. As shown in Fig.3, we observed a significant positive correlation between dystonic symptom severity and low-frequency power in the GPi as the main output nucleus of the basal ganglia ($R=0.89$, $P=.0014$), but not the striatum as their main input nucleus nor the GPe. In addition to low-frequency power in the GPi, the coupling strength as measured by imaginary part of coherence in the low-frequency band between striatum and GPi – which presumably reflects direct pathway activity - correlated significantly with symptom severity ($R=0.74$, $P=.009$), see Fig. 3F. In contrast, low-frequency coupling strength between striatum and GPe – as possible correlate of indirect pathway activity - did not ($R=-0.002$, $P=0.13$, Fig. 3F), nor did coupling strength between GPi and GPe ($R=0.6$, $P=0.05$).”

Figure 3: Connectivity patterns across basal ganglia structures and their correlation to symptom severity. (A) Shown are averaged coherence spectra between striatum and GPe (purple), striatum and GPe (blue), GPe and GPi (green) and averaged shuffled coherence (grey) across hemispheres. Shaded areas indicated standard deviation of the mean. (B) All recorded basal ganglia structures are functionally coupled in the low-frequency band as shown by significantly higher coherence values when compared to shuffled data. Black dots indicate mean value per hemisphere. In box plots, central marks indicate the median and edges the 25th and 75th percentiles of the distribution. (C) Shown are averaged spectra of granger causality with GPi (blue) or striatum (purple) as source. (D) When each frequency bin is considered separately, the majority of the low-frequency band is led by the striatum in 63% and by the GPi in 37% of cases. (E) Dystonic symptom severity correlates with averaged low-frequency power in the GPi, but not the striatum. (F) Moreover, the significant coupling strength of low-frequency activity between the striatum and the GPi, but not between striatum and GPe hints towards an involvement of the direct but not the indirect pathway. ** $p < 0.01$.

Reviewer #2 (Remarks to the Author):

The paper presented exploratory investigations into the direct and indirect pathways of the basal ganglia circuits involved in the pathophysiology of dystonia. It is unique and intriguing although there might be some uncertainties to be further clarified in the future. One suggestion is to provide details on the DBS targeting and implantation procedures in supplement. The location of all electrodes looks too good to be true. Additional information on the procedure and original MRI scanning would further consolidate the results as LeadDBS doesn't always work well.

We thank the reviewer for this helpful suggestion. We have now prepared a dedicated supplement including further information on the surgical planning and MRI acquisition as well as coronal sections of individual MRIs superimposed with CT artifacts and markings for the location of the bipolar recording sites. Despite this, we believe that perhaps it should be highlighted again that suboptimal electrode localization in either one of the three structures (Striatum, GPe, Gpi) was an exclusion criterium of our study. Since the start of the recruitment in 2017, we recorded intracranial activity in 19 patients with dystonia, but only 10 patients were included in our study, as they fulfilled our strict criteria on electrode localization. We hope that this clarification alongside the new supplement satisfies the expert reviewer. We added this information to the revised version of the manuscript which now reads as follows:

“Of 19 patients with dystonia, 10 patients fulfilled these inclusion criteria and were thus considered for further analysis of the neurophysiological data. In patient #7, no contact of the left DBS-electrode was localized in the striatum and the left hemisphere was thus excluded, resulting in 19 hemispheres included in the present study. In one case (patient # 6, right hemisphere), one DBS-contact of the GPi-recording was placed slightly ventral to the GPi, in adjacent white matter. Primary DBS target was the posteroventral lateral, motor portion of the GPi. Further details on surgical planning as well as electrode localization in individual MRI-sequences can be found in the supplementary material, supplementary table 3 and supplementary Figure 6.”

“Supplementary material

Surgical procedure

All patients underwent bilateral DBS in the globus pallidus internus (GPi) at Charité University Hospital, Berlin (n=7) or Medizinische Hochschule Hannover (n=3). Electrodes were targeted at the posteroventrolateral

portion of the GPi. Permanent octopolar DBS leads (Boston Scientific Vercise) were implanted under general anaesthesia in all patients. The stereotactic target points were determined by a combined approach of indirect standard coordinates followed by direct refinement according to individual anatomy of nuclei as assessed by structural preoperative MRI (see below). The preliminary target for GPi (reflecting the center of the most distal electrode contact) was identified 3-5 mm anterior, 19-21 mm lateral and 3-4 mm below the midcommissural point. Under these circumstances, with a trajectory that is angled by 70–80° in the sagittal plane and by 75–85° in the coronal plane, the more rostral contact pairs are likely to lie in the globus pallidus externus (GPe) and Putamen. Intraoperatively, the final electrode position was verified using microelectrode recordings and macrostimulation with tetanic stimulation to assess proximity to the internal capsule.

Preoperative MRI acquisition

In all patients, high-resolution T1w and T2w MRI scans were obtained preoperatively using a 3.0T clinical MRI scanner (Skyra or Vida Magnetom, Siemens, Erlangen, Germany). In addition, fast gray matter acquisition T1 inversion recovery (FGATIR) sequences were acquired in $n = 7$ patients. The complete MRI acquisition protocol consisted of a 3-plane localizing scout, a T1w 3-dimensional (3D) magnetization-prepared rapid acquisition gradient echo sequence, a T2w turbo spin echo (TSE) sequence, and a T1w 3D FGATIR sequence. A detailed overview of the acquisition parameters used for the protocol can be obtained from Supplementary table 3. Individual MRI imaging with FGATIR sequences allow a refined targeting to the transition between the middle and dorsal third of the GPi as depicted on axial FGATIR sequences.”

Supplementary table 3. Imaging Sequences and Acquisition Parameters Employed During Preoperative MRI

Parameter	T1w 3D MP-RAGE	T2-TSE	T1w 3D FGATIR
Repetition time, ms	2,300	13,320	3,000
Echo time, ms	2.32	101	3.44
Inversion time, ms	900	n.a.	414
Inversion pulse angle	90°	n.a.	180°
Field of view, mm	240 × 240	250 × 250	240 × 240
Slices, mm	192 × 0.9	70 × 2.0	160 × 1
Orientation	Sagittal	Axial	Axial
Bandwidth, Hz/Px	200	217	130
Acquisition time, min	5:21 min	4:15 min	6:17 min
Voxel size, mm ³	0.9 × 0.9 × 0.9	0.7 × 0.7 × 2.0	0.9 × 0.9 × 1.0

Note that sequences were obtained on a 3.0T clinical MRI scanner (Skyra or Vida Magnetom, Siemens, Erlangen, Germany), and parameters were optimized accordingly. FGATIR can be implemented on 1.5T scanners; however,

parameters would have to be adjusted accordingly. 3D = 3 dimensional; FGATIR = fast gray matter acquisition T1 inversion recovery; MP-RAGE = magnetization-prepared rapid acquisition gradient echo; n.a. = not applicable; Px = pixel; T1w = T1-weighted; TSE = turbo spin echo.

Supplementary figure 4. Overlay of preoperative MRI (T1) and postoperative CT with each contact-pair marked in native space. Shown are coronal planes in which the DBS-artifact and the recording site of the respective basal ganglia is visible (recording site is visualized with a radius of 4 mm in GPI=green, Gpe=blue, striatum=purple), separately for each patient. The size of the visualized recording site may vary according to the angle of the chosen MRI frame that best displays the electrode artefact. In subject #1, #6 and #8, the left and right DBS-electrode are not visible on the same coronal plane. In these cases, the coronal planes for each hemisphere have been merged for visualization purposes. Note that for patient #7 only the right hemisphere is shown, because the left DBS-electrode did not cover the striatum and was thus excluded from further analysis.

REVIEWERS' COMMENTS

Reviewer #2 (Remarks to the Author):

The authors have well addressed all of the questions.
It is a nice paper to read.